# Antimicrobial Properties on Non-Antibiotic Drugs in the Era of Increased Bacterial Resistance

**DOI:** 10.3390/antibiotics9030107

**Published:** 2020-03-02

**Authors:** Maria Lagadinou, Maria Octavia Onisor, Athanasios Rigas, Daniel-Vasile Musetescu, Despoina Gkentzi, Stelios F. Assimakopoulos, George Panos, Markos Marangos

**Affiliations:** 1Emergency Department, University General Hospital of Patras, 26504 Patras, Greece; 2Department of Infectious Diseases, University General Hospital of Patras, 26504 Patras, Greece; sassim@upatras.gr (S.F.A.); george.panos@upatras.gr (G.P.); mmarangos@yahoo.com (M.M.); 3Medical Department, University of Brasov, 500036 Transilvania, Romania; mariaonisor@gmail.com (M.O.O.); musetescu.daniel@gmail.com (D.-V.M.); 4Medical Department, University of Patras, 26504 Patras, Greece; athrigas@hotmail.com; 5Department of Paediatrics, University General Hospital of Patras, 26504 Patras, Greece; gkentzid@hotmail.com; 6Department of Internal Medicine, University General Hospital of Patras, 26504 Patras, Greece

**Keywords:** antimicrobial agents, bacteria, NSAIDs, antidepressants, opioids, statins, antihistamines, non antibiotics

## Abstract

In recent years, due to the dramatic increase in and global spread of bacterial resistance to a number of commonly used antibacterial agents, many studies have been directed at investigating drugs whose primary therapeutic purpose is not antimicrobial action. In an era where it is becoming increasingly difficult to find new antimicrobial drugs, it is important to understand these antimicrobial effects and their potential clinical implications. Numerous studies report the antibacterial activity of non-steroidal anti-inflammatory drugs, local anaesthetics, phenothiazines such as chlorpromazine, levomepromazine, promethazine, trifluoperazine, methdilazine and thioridazine, antidepressants, antiplatelets and statins. Several studies have explored a possible protective effect of statins inreducing the morbidity and mortality of many infectious diseases. Various non-antibiotic agents exhibit antimicrobial activity via multiple and different mechanisms of action. Further studies are required in the field to further investigate these antimicrobial properties in different populations. This is of paramount importance in the antimicrobial resistance era, where clinicians have limited therapeutic options to combat problematic infections.

## 1. Introduction

The increase in antimicrobial resistance is a major health problem worldwide. Recent discoveries of plasmid-transferable genes that mediate resistance to carbapenems and colistin indicate that the last defensive wall against multi-drug-resistant pathogens has already been breached [1]. In recent years, due to the dramatic increase in and global spread of bacterial resistance to a number of commonly used antibacterial agents, many studies have been directed at investigating drugs whose primary therapeutic purpose is not antimicrobial action [2]. Drug classes such as neuroleptics, antihistamines, antidepressants, antiplatelets and non-steroidal anti-inflammatory drugs (NSAIDs) have a greater or lesser degree of broad-spectrum antibacterial activity [2]. These non-antibiotic drugs act in multiple and different ways on microbial growth. They may have direct antimicrobial activity (antimicrobial non antibiotics), increase the efficacy of an antibiotic when co-administered (helper compounds), or change the pathogenicity of microorganisms or the activity on the physiology, such as modulating macrophage activity (Table 1) [3].

In an era where it is becoming increasingly difficult to find new antimicrobial drugs, it is important to understand these antimicrobial effects and their potential clinical implications [4]. After searching in the literature, we found the most recent articles referring to that topic. The purpose of this narrative review is to summarize the latest literature in the field.

## 2. NSAIDs

Non-steroidal anti-inflammatory drugs are among the most widely used in the world. Acetaminophen, acetylsalic and other non-steroidal anti-inflammatory drugs (NSAIDs), such as diclofenac and ibuprofen, are some of the most commonly used drugs. In addition to their anti-inflammatory action, they have analgesic and antipyretic properties. It has also been shown, but neglected, for over 20 years, that NSAIDs do have direct and indirect antimicrobial effects [4]. Their main mechanism of action involves the inhibition of cyclooxygenase, which leads to a decrease in the synthesis of prostaglandins [5].

NSAIDs at therapeutic or higher levels can be active against bacteria, viruses and fungi. At therapeutic plasma levels, acetylsalicylic acid (ASA) and salicylic acid (SAL) inhibit the growth of *Campylobacter pylori*, *Helicobacter pylori*, and *Klebsiella pneumoniae*, as well as *Epidermophyton floccosum*, *Microsporum* spp., and *Trichophyton* spp. SAL is also active against *Staphylococcus aureus* by reducing fibronectin binding. At slightly higher concentrations. above the therapeutic levels, SAL reduces flagellin production in *E. coli*, as well as flagellum production in *P. mirabilis*. Moreover, Ibuprofen can inhibit the growth of *Escherichia coli*, because, at a very low concentration (0.002 mg/mL), it decreases the adhesion of *E. coli* to uroepithelial cells. This happens from reduced fimbria production, as well as from changes in surface hydrophobicity influencing the interaction between bacterial and host cells. ASA/SAL inhibits the replication of some viruses, such as hepatitis C virus, flavivirus and influenza virus. One of the potent mechanisms is inhibition of the transcription factor nuclear factor-kappa B (NF-kB). NF-kB plays a crucial role in the expression of multiple cellular and viral genes involved in inflammation, including interleukin-1 (IL-1), IL-6, and adhesion molecules. Another mechanism is the activation of p38 mitogen-activated protein kinase and mitogen-activated protein kinase/extracellular signal-regulated kinase ½ [4].

Ibuprofen, in addition to its main function, inhibits the growth of *Escherichiacoli* at therapeutic levels and, at low pH, also *Staphylococcus aureus*, *Microsporum* spp., and *Trichophyton* spp. Diclofenac showed invitroantimicrobial activity against different bacterial pathogens, including *Escherichia coli*. In an interesting approach, Alqahtani andcolleagues, in 2018, prepared chitosan nanoparticles loaded with diclofenac and demonstrated high activity of this cluster against *Staph. aureus* and *B. subtilis*, which depended on the molecular weight of chitosan and pH.Ibuprofen and diclofenac have significantly more pronounced antibacterial activity against *Enterococcus faecalis* in comparison with Ca(OH)_2_. The antibacterial activity of ibuprofen and diclofenac against *Enterococcus faecalis* is weaker than that of antibiotics (amoxicillin and gentamycin) [6].

NSAIDs have a pleiotropic action: either they alter the expression of many genes (in *Escherichia coli*, more than 144 genes; in *Ps. aeruginosa*, more than 331 genes), or they reduce the production of some molecules (polysaccharide capsule of *Klebsiella pneumonia* and hemolysin, elastase, protease and pyocyanin of *Pseud. aeruginosa*, hemolysin and teichoic acid of *Staphylococcus aureus* and *epidermidis* resprectively. In *H. pylori*, ASA and indomethacin reduce urease and vacuolating cytotoxin activities [4]. The mechanism of action of diclofenac seems to reside mainly in the inhibition of DNA synthesis. Particularly in *Escherichia coli*, some NSAIDs showed an ability to inhibit the DNA polymerase III β subunit. The change in this subunit as a consequence of the binding of the NSAID molecule results in the inhibition of DNA replication and repair.

In mice, ASA and ibuprofen enhance the effect of pyrazinamide during the initial phase of tuberculosis treatment, and diclofenac protects mice from dying from *Salmonella* infection. In rabbits with *Staphylococcus aureus* endocarditis, ASA reduces vegetation bacterial density, hematogenous bacterial dissemination and the frequency of embolic events [4].

## 3. Local Anaesthetics

Since the introduction of cocaine in 1884, local anaesthetics (LA) have been used as a major agent of pain relief. However, numerous studies in recent years have shown the potential role of local anaesthetics as antimicrobial agents. Their supplemental role is in the potential prevention and treatment of surgical site infections [7].

Lidocaine, which is the most studied agent, has been proved to have in vitro antibacterial effects on common bacterial pathogens which are isolated from nosocomial wound infections—*Enterococcus faecalis*, *Escherichia coli*, *Ps. aeruginosa*, and *Staphylococcus aureus*—as well as a number of resistant strains of methicillin-resistant *Staphylococcus aureus* and vancomycin-resistant *Enterococci*. Results have shown a concentration-dependent activity against Gram-positive and Gram-negative bacteria, which are among the main pathogens of surgical infections. *Escherichia coli*, *Ps. aeruginosa* and *Staphylococcus aureus* were found to be less sensitive to lidocaine [8]. *Candida albicans* also belongs to the antimicrobial spectrum of lidocaine [3]. The antimicrobial activity of bupivacainehas also been recently reviewed. Its antimicrobial activity against coagulase-positive *Staphylococcus*, followed by coagulase-negative *Staphylococcus*, *Klebsiella*, *Enterobacter*, *Esherichia coli*, and *Proteus spp* [8,9], has been evaluated.Bupivacaine at concentrations used in epidural anaesthesia inhibited the growth of both gram(+) and gram(−) microorganisms: *Escherichia coli*, *Staphylococcus aureus*, *methicillin resistant Staphylococcus aureus*, *Staphylococcus epidermidis*, *Staph. pyogenes*, *Enterococcus faecalis* and *Strep. pneumoniae*. Bupivacaine (0.5%) also has been proven to have an inhibitory effect on the growth of *Candida albicans* [10]. Ropivacaine is another amide LA. It has been reported that, in higher concentrations (10 mg/mL), it inhibited the growth of both *Escherichia coli* and *Staphylococcus aureus* [8].

Limited studies exist regarding the mechanisms of action of LA antimicrobial activity. Some of them include the disruption of the bacterial cell membrane, inhibition of cell wall synthesis, dysfunction of cellular respiration, alteration in DNA synthesis, lysis of protoplasts, alteration in permeability and leakage of intracellular components, ultrastructural changes and inhibition of membrane-bound enzymatic activities [8,9].

## 4. Opioids

After the discovery of the antibacterial effect of local anaesthetics, it was presumed that certain opioids might inhibit the growth of microorganisms, since some of them have an anaesthetic effect as well. It appears that the antibacterial effect might be related to their anaesthetic capacity.

Tramadol is a synthetic opioid, an analogue of codeine used to treat patients with moderate to moderately severe pain. Tamanai-Shacoori [11] and colleagues tested tramadol in cultures of *Escherichia coli* and *Staphylococcus aureus* at two different concentrations. Tramadol at a concentration of 12.5 mg/mL had an inhibitory effect by reducing the *Escherichia coli* and *Staphylococcus aureus* growth, respectively, whereas, at 25 mg/mL, the inhibitory effect was increased for *Staphylococcus aureus*, but tramadol induced a bactericidal effect for *Escherichia coli* [12]. In another study, more strains were used—*Escherichia coli*, *Staphylococcus aureus*, *Staphylococcus epidermidis*, and *Ps. aeruginosa*—with similar results. Tramadol has been shown to be active against *Escherichia coli*, *Staph. Epidermidis*, at concentrations around 25 mg/mL for 6 h, or around 12.5 mg/mL for 24 h [11]. It is not as effective against *Staphylococcus aureus* and *Ps. Aeruginosa*, even at such high concentrations. It should be acknowledged that Tramadol has dose- and time-dependent bactericidal activity against the above-mentioned bacteria [11].

In the 20th century, meperidine and/or pethidine are the drugs of choice among the opioids in the management of acute pain by most physicians. Regarding their antimicrobial activity, meperidine had the highest antibacterial effect in a study which assessed the antibacterial activity of morphine, meperidine and fentanyl against strains of coagulase-negative *Staphylococcus*, *coagulase positive staphylococcus*, *E. coli*, *Klebsiella pneumonia*, *Enterobacter*, *Ps. aeruginosa* and *Proteus* [13]. According to another study, meperidine inhibited the growth of *Str. pneumoniae* at a MIC = 25 mg/mL, whereas *S. pyogenes*, *MRSA*, *Staph. epidermidis* and *Escherichia coli* at a MIC = 5 mg/mL *Staphylococcus aureus* was inhibited only at two and a half times epidural concentration [14].

Methadone is a synthetic, long-acting opioid with pharmacologic actions qualitatively similar to morphine, and is active by oral and parenteral routes of administration [15]. The effect of methadone was evaluated alone and in combination with other antibiotics on strains of *Staphylococcus aureus*, *Ps. aeruginosa* and *Serratia marcescens.* The results showed that methadone has a bigger antibacterial effect against *Staphylococcus aureus* than against the two other Gram-negative organisms tested, but only at supraphysiological concentrations. Moreover, it has a partial or total synergistic effect with other antimicrobial agents like nafcillin or gentamicin [16].

## 5. Antipsychotics

According to Nehme et al., none of the investigated atypical antipsychotics (aripiprazole, clozapine, olanzapine, quetiapine, risperidone, reserpine, haloperidol and sulpiride) showed antibacterial activity against tested bacterial strains. Only phenothiazines (chlorpromazine, thioridazine, fluphenazine) and thioxanthenes (flupenthixol, chlorprothixene) showed antibacterial activity [17].

Poulsen et al. [18], showed that phenothiazines exhibit antibacterial activity against Gram-positive cocci, *Mycobacteria* and some Gram-negative rods such as *Shigella spp.* (MIC 20–30 μg/mL), *Escherichia coli* and *Salmonella spp.* (MIC 100 μg/mL) [18,19]. Numerous studies report the antibacterial activity of phenothiazines such as chlorpromazine, levomepromazine, promethazine, trifluoperazine, methdilazine and thioridazine against strains of *Mycobacterium tuberculosis*, with MICs that vary between different studies. *Staph. aureus* strains were more susceptible to these agents (MIC 32–64 μg/mL), whereas Gram-negative strains of *Escherichia coli*, *Acinetobacter baumani* and *Klebsiella pneumoniae* were generally more resistant (MIC 64–128 μg/mL). Concentrations needed for these activities greatly exceed the highest achievable level in plasma (0.5 μg/mL). [19] Consequently, these compounds cannot stand as antibacterials on their own. However, they’re able to enhance the activity of antibiotic agents [17,19]. Therefore, they can be used as adjuvants for reducing the doses of a given antibiotic or reversing bacterial resistance to them [20]. More specifically, thioridazine (TZ) is a typical antipsychotic agent which belongs to phenothiazine drug group and it has been previously shown that the racemic mixture of thioridazine (TZR, consisting of +ΤΖ and −TZ enantiomers) sensitizes *Staph. aureus* to classical antibiotics. Combinations of TZR and −TZ with dicloxacillin are equally effective against MSSA and MRSA as well [19].

Phenothiazines are known to have in vitro activity against *Plasmodium falciparum* and are active against *Leishmania* spp, in concentrations they are either toxic or clinically non effective. Moreover they are active against a wide range of viruses (*Herpes simplex, Tulip Breaking Virus, HBV, Measles Influenza, Sinbis and Vesicular Stomatitis Virus, SV40, Arenavirus, HIV, Human Herpes Virus, JC virus*) via inhibition of the binding of the virus to host cell membrane receptors, affecting the calcium-dependent endocytosis of viruses and viral DNA replicatiοn [20].

Phenothiazines act by reducing the adherence of such pathogens in the endothelial cells. In addition, phenothiazines inhibit the activity of calcium-dependent ATPase, resulting in the acidification of phagolysosomes, the activation of hydrolases and, eventually, leading to the inhibition of the replication of bacterium [21]. Phenothiazines have been shown to inhibit the efflux pumps that account for antibiotic resistance in bacteria, such as the Nor A efflux pump of *Staphylococcus aureus* and the QAC efflux pump of plasmid-carrying, multidrug-resistant *Staphylococcus aureus*, the AcrAB efflux pump of *Escherichia coli* and others [20,21]. They achieve that by inhibiting the binding of calcium to the calcium-dependent ATPase of plasma membrane, the activity of which provides the energy required for the transport [20].

## 6. Antidepressants

Antidepressants represent a family of substances primarily used for psycho-pharmaceutical purposes in the treatment of sentimental disorders. In recent years, their antibacterial effects have been exhibited in numerousstudies. They have gained considerable interest due to the need for new chemotherapeutics against increasingly drug-resistant microbial infections. Among antidepressants, the most prominent group for their antimicrobial actions are the Serotonin Reuptake Inhibitors (SSRIs) and tricyclic antidepressants (especially amitriptyline).

The antimicrobial effect of the SSRIs sertraline, fluoxetine and paroxetine are especially active against Gram-positive bacteria such as *Staphylococcus* and *Enterococcus* [21,22,23], although they are not active against others: *S. pneumoniae*, *S. pyogenes*, *S. agalactiae* [21]. In addition, SSRIs present good activity against potentially toxigenic *Enterobacteria*, such as *Citrobacter spp*, *Ps. aeruginosa*, *Kl. pneumoniae* and *Morganella morganii* [23,24]. Moreover, they exhibit surprising activity against *H. influenza*, *Moraxella catarrhalis*, *Campylobacter jejuni*, and even against *Acinetobacter* [21]. Additionally, they are active against some anaerobes. such as *Bacteroides fragilis* [21,23], *Clostridium perfringens* and *Clostridium difficile* [23,24]. In the study of Muhammad et al., among the tested SSRIs (sertraline, citalopram, venlafaxine), sertraline was the most potent, exhibiting intrinsic antibacterial activity against strains of *Staphylococcus aureus*, *Escherichia coli* and *Ps. aeruginosa*, as well as antifungal activity against strains of *Aspergillus* and *Fusarium* [10,19]. Lass-Flörl et al. investigated the fungicidal activity of SSRIs (fluoxetine, seroxate, paroxetine, sertraline and reboxetine) against *Aspergillus spp.* (*A. fumigatus*, *A. flavus*, *A. terreus*) and *Candida parapsilosis* [21,23].

Amitriptyline hydrochloride exhibits significant inhibitory action on Staphylococcus spp., Bacillus spp., Vibrio Cholerae, Micrococcus spp., Lactobacilus sporogenes and Citrobacter spp., and moderate inhibitory action on Shigella, Salmonella, Vibrio parahaemolyticus, Escherichia coli, Kl. pneumonia, and Pseudomonas spp. [24]. It also inhibits the growth of fungi, exhibiting good antifungal activity against Cryptococcus spp. (500 μg/mL) and moderate activity against Candida albicans [24]. In vivo experiments exhibited significant protection on mice at 25 and 30 μg/mL body weight, showing no toxicity at these doses [24].

The proposed mechanism explaining the antimicrobial action of SSRIs is the inhibition of efflux pumps. This effect may explain the synergistic effect of SSRIs in combination with certain antibiotics, as well as their effects against some strains of bacteria resistant to antimicrobial agents [25,26], confirmed by decreases in the minimum inhibitory concentrations (MICs) of antibiotics when combined with antidepressants [23,27]. Muhammad et al. showed that the antibacterial activity of all tested antibiotics was significantly increased with the addition of increasing concentrations of sertraline [19,27]. The synergistic effects were observed by the combination of some SSRIs and antibiotics against microorganisms, including those which are resistant, such as *Corynebacterium urealyticum* [21,23,27,28]. They increase the activity of some antibiotics, such as tetracyclines and fluoroquinolones [21]. They also present antiplasmid activity, targeting the replicating plasmid DNA and the DNA gyrase enzyme. Besides this, tricyclic antidepressants presented in vitro activity against *Plasmodium falciparum* [19,29,30] and *Leishmania spp* [31].

## 7. Antiplatelets Drugs

Among the antiplatelets medications, ticagrelor is the drug that showed the most potential antibacterial activity. Ticagrelor, which is a P2Y12 receptor antagonist, is unique among antiplatelet drugs not only because it inhibits the platelet P2Y12 receptor in a reversible manner, but also because it demonstrates a wide palette of advantageous pleiotropic effects [30,32].

Lancellotti et al. [33] studied the antimicrobial effect of ticagrelor on nine different bacterial strains (including both Gram negative and positive ones) The Gram-positive bacteria used were: *methicillin-resistant Staphylococcus epidermidis (MRSE); methicillin-sensitive Staphylococcus aureus (MSSA); methicillin-resistant Staphylococcus aureus (MRSA); glycopeptide intermediate S. aureus (GISA); Enterococcus faecalis; vancomycin-resistant E. faecalis (VRE); Streptococcus agalactiae* and Gram-negative bacteria were: *Escherichia coli and Pseudomonas aeruginosa.* The results showed that ticagrelor had bactericidal activity on all the Gram-positive tested strains. The minimum bactericidal concentrations established for each of the bacteria were: 20 μg/mL for *MSSA*, *MRSA*, *GISA*, *VRE*, 30 μg/mL for *MRSE*, 40 μg/mL for *E. faecalis* and *S. Agalactiae* [32,33]. Moreover, it has been concluded that the values are higher in contrast to vancomycin, with the rapid killing of late-exponential phase cultures of *MRSA* [33]. It has also been shown that twenty-four hours after the inoculation, the ability to kill *MRSE* and *VRE* is similar to daptomycin [33]. Regardless of the promising results obtained for Gram-positive bacteria, ticagrelor was not able to provide any antibacterial activity against Gram-negative strains in concentrations up to 80 μg/mL [33]. Moreover, it has been concluded that values are higher in contrast to vancomycin, with the rapid killing of late-exponential phase cultures of *MRSA* [33]. It has also been shown that twenty-four hours after the inoculation, the ability to kill *MRSE* and *VRE* is similar to daptomycin [33]. Regardless of the promising results obtained for Gram-positive bacteria, ticagrelor was not able to provide any antibacterial activity against Gram-negative strains in concentrations up to 80 μg/mL [33].

A synergism between the antiplaletets and different antibiotics has also been reported. A subminimal bactericidal concentration of ticagrelor (10 μg/mL) combined with vancomycin (4 μg/mL) killed approximately 50% of the initial MRSA inoculums [33]. Ticagrelor has increased the bactericidal activity of rifampicin, ciprofloxacin, and vancomycin in a disk-diffusion assay [33]. In the Targeting Platelet-Leukocyte Aggregates in Pneumonia with Ticagrelor (XANTHIPPE) research, patients who received treatment with ticagrelor had improved lung function during sepsis and pneumonia [33]. Likewise, the Platelet Inhibition and Patient Outcomes (PLATO) trial demonstrated that the mortality risk after pulmonary adverse events and sepsis in acute coronary syndrome appears to be lower when using ticagrelor as therapy in comparison to clopidogrel [34].

## 8. Antihistamines

Antihistamines have had a pioneering role in the treatment of allergic diseases for many years. However, recent studies have shown that antihistamines may exhibit a different rolein medicine, which is to provide an antibacterial effect against microorganisms.

Azelastine has proven a great ability to provide bactericidal activity on the following Gram-positive bacteria: *Staphylococcus aureus* and *Staphylococcus epidermidis*, *Enterococcus faecium* [35]. While the antihistamine hasa moderate effect on *Escherichia coli* and *Klebsiella spp*, there are no results on *Ps. Aeruginosa* [35]. The post-antimicrobial effect of azelastine was studied against *Staphylococcus aureus* and *E. coli* isolates [35]. The post-exposure effect was sustained for 3.36 h against the tested *Staphylococcus aureus* isolate compared to only half an hour for the *Escherichia coli* isolate [35]. This information is highly essential, as it may be a reason to reduce the dose administered and prolong the time interval of administration. This will decrease any possible adverse effects [35].

Cetirizine, when tested for antimicrobial activity against 42 bacterial strains belonging to the Gram-positive and -negative groups, it was proven that a majority of them were inhibited at 200–2000 µg/mL concentration, and a few were also susceptible below 200 µg/mL concentration [36]. The order of sensitivity towards cetitrizine was: *Bacillus sp*, *Vibrio cholera*, *Staphylococcus aureus*, *Escherichia coli* and *Shigella spp* [36]. Moreover, cetirizine in vitro is a bactericidal agent for *Staphylococcus aureus* and *Salmonella typhi*. On the other hand, during the in vivo study, cetirizine gave significant protection to the challenged mice with *S. typhimurium* at a high (100 μg/mL) concentration [36]. El-Nakeeb, M.A. et al. concluded that cetirizine possessed a slight bacteriostatic activity in relatively higher concentrations against both tested Gram-positive and Gram-negative bacteria. However, it lacked activity against the *Ps. aeruginosa* and *Pr. mirabilis* isolates in the tested concentration range [35].

Terfenadine is another promising antihistamine which possesses antimicrobial activity. The MIC values of terfenadine and its analogues were 16, 4, and 2 μg/mL against methicillin-resistant (MRSA), vancomycin intermediate (VISA) and vancomycin-resistant (VRSA) *S. aureus*, respectively [37]. It has also demonstrated antimicrobial activity toward *Enterococcus faecium*, *Enterococcus faecalis*, and *M. tuberculosis*. Neither terfenadine nor its analogues have been active against wild-type Gram-negative species to date. The MIC is >256 μg/mL for both *Klebsiella pneumoniae* and *Escherichia coli.* The MIC value for *Acinetobacter baumannii* is 256 μg/mL as well [37]. Mepyramine, a histamine H1 receptor antagonist, enhances the efficacy of antibacterials against *Escherichia coli* [38]. Promethazine and cyproheptadine showed antibacterial activity against the tested *K. pneumoniae* isolates, with MIC values ranging from 400 to 1000 µg/mL (far more than their biological levels) [39].

The mechanism of action which enables the antihistamines to have antibacterial properties has not been completely identified, although there are several concepts. Firstly, the alteration inmembrane permeability is one of the major mechanisms underlying the antibacterial effects of antihistamines against the tested bacteria [35]. Moreover, absorption of the antihistamines onto the bacterial cell surface may help to generate an antibacterial effect [40]. Another hypothesis pertains to the expansion of the hydrophobicity which, in turn, would increase the surface activity [41]. As a result, an elevated antimicrobial response could be due to surface action and increased hydrophobic character [42]. Other researchers, however, propose a theory that emphasizes the role of bacterial efflux pumps. Their inhibition by antihistamines augments the efficacy as an antibacterial substance [39]. Apart from that, Bakker et al. suggested a mechanism that involves bacterial type II topoisomerase inhibitors targeting both DNA gyrase and topoisomerase IV [43].

## 9. Statins

Statins are important lipid-lowering agents which reduce heart-associated morbidity and mortality. Except for their action as lipid-reduction agents, they have anti-inflammatory and immunomodulatory activities too. Several studies have explored a possible protective effect of statins to reduce the morbidity and mortality of many infectious diseases [44].

Statins, especially atorvastatin and simvastatin, have been studied for their antibacterial activity. Many studies have demonstrated a bigger antimicrobial effect of atorvastatin and simvastatin compared to rosuvastatin against *Methicillin-sensitive staphylococcus aureus (MSSA), Methicillin-resistant Staphylococcus Aureus (MRSA), Vancomycin-Susceptible Enterococci (VSE), Vancomycin-Resistant Enterococcus (VRE), Acinetobacter baumannii, Staphylococcus epidermidis*, and *Enterobacter aerogenes*. Furthermore, atorvastatin has been proven to be more active against *Escherichia coli*, *Proteus mirabilis* and *Enterobacter cloacae* compared to both simvastatin and rosuvastatin [45].

Statins induce their hypolipidemic action via inhibition of HMG-CoA reductase. In bacterial cells, HMG-CoA reductase is a basic component in the biosynthesis of isoprenes [45]. Thus, it is unlikely that the antibacterial activity of statins can be attributed to the known mechanism of action (i.e., the inhibition of HMG-CoA reductase). Other possible mechanisms might be related to the cytotoxic capacity that characterizes some statins, especially atorvastatin and simvastatin. It has been proven that statins suppress cells growth and promote apoptosis. It is possible that the currently reported antibacterial activity of statins is related to such effects [45]. It should be noted that these agents, in particular simvastatin, have such an antimicrobial effect in vitro at concentrations greater than those commonly used. Therefore, statins probably do not exert a significant antimicrobial effect in patients, but these data have revealed an unanticipated class effect and further testing of statins and their metabolites is warranted [46].

In conclusion, we have shown that various non antibiotic agents exhibit antimicrobial activity via multiple and different mechanisms of action. Further studies are required in the field to further investigate these antimicrobial properties in different populations. This is of paramount importance in the antimicrobial resistance era, where clinicians have limited therapeutic options to combat problematic infections.

**Table 1 antibiotics-09-00107-t001:** Non-antibiotic agents with antimicrobial activity, their antimicrobial spectrum and approved or potential mechanisms of action.

Non Antibiotic Agent	Antimicrobial Spectrum	Suggested Mechanisms of Action	References
NSAIDs			
Acetysalicylic acid/salicylic acid	*Staphylococcus aureus, Ps. aeruginosa, Campylobacter pylori, Helicobacter pylori, Epidermophyton floccosum, Microsporum spp., Trichophyton spp. Hepatitis C virus, influenza virus*	Modulation of the expression of many genes, reduction of polysacharide capsule production, in *S. aureus* reduction of hemolysin production, in *Ps. aeruginosa* reduction of elastase, hemolysin, protease pyocyanin production, in *H. pylori* reduction in hemolysin production	[4]
Ibuprofen, diclofenac, fulbiprofen	*Escherichia coli, Staphylococcus aureus, Microsporum spp., and Trichophyton spp., E. faecalis*	Inhibition of DNA synthesis. In *E. coli*: inhibition of the DNA polymerase III β subunit	[6]
**Local anaesthtetics:**			
Lidocaine, bupivacaine	*Escherichia coli, Ps. aeruginosa, Staphylococcus aureus, Candida albicans*, *Coagulase(-) Staphylococcus, MRSA, Klebsiella, Enterobacter, E. coli, E. Faecalis and Proteus species*	Disruption of bacterial cell membrane, inhibition of cell wall synthesis, dysfunction of cellular respiration, alteration in DNA synthesis, lysis of protoplasts, alteration in permeability and leakage of intracellular components, ultrastructural changes, inhibition of membrane-bound enzymatic activities	[8,9]
**Opioids**			
Tramandol, meperidine	*Escherichia coli, S. aureus* *S. epidermidis, Ps. aeruginosa* *S. pneumoniae, S. pyogenes, MRSA*	No data exist on possible mechanism	[11,13]
**Antipsychotics:**			
Phenothiazines(chlorpromazine, thioridazine, fluphenazin)	Gram(+)cocci, *Mycobacteria*, *Shigella spp., E. coli, Salmonella spp., Acinetobacter baumanii, Klebsiella pneumonia, Plasmodium falciparum, herpes simplex, HBV, measles influenza, SV40, HIV, human herpes virus, JC virus*	Adherence reduction in the pathogens in the endothelial cells, efflux pumps inhibition. Inhibition of: the binding of virus to host cell membrane receptors, the calcium-dependent endocytosis of viruses and the viral DNA replicatiοn	[20,21]
**Antidepressants:**			
Sertraline, fluoxetine, paroxetine	*Staphylococcus spp, Enterococcus, Citrobacter spp, Ps. aeruginosa, Klebsiella pneumoniae and Morganella morganii, Haemophilus influenza, Moraxella catarrhalis, Campylobacter jejuni, Acinetobacter, Bacteroides fragilis, Clostridium perfringens and Clostridium difficile*	Inhibition of efflux pumps, antiplasmid activity targeting the replicating plasmid DNA and the DNA gyrase enzyme.	[25,26]
Amitriptyline hydrochloride	*Aspergillus and Fusarim, Aspergillus spp, Candida parapsilosis, Plasmodium falciparum, Leishmania spp*, *Staphylococcus spp., Bacillus spp., Vibrio Cholerae, Micrococcus spp., Lactobacilus sporogenes, Citrobacter spp., Shigella, Salmonella, V. parahaemolyticus, E. coli, K. pneumonia, Pseudomonas spp*		
**Antiplatelets**			
ticagrelor	*MRSE, MSSA, MRSA, glycopeptide intermediate Staphylococcus aureus (GISA); Enterococcus faecalis; vancomycin-resistant E. faecalis (VRE); Streptococcus agalactiae*	No data exist on possible mechanism of action	[32,33]
**Antihistamines**			
Azelastine	*Staphylococcus aureus, S. epidermidis, Enterococcus faecium, Escherichia coli, Klebsiella spp*	Alteration of membrane permeability Absorption of the antihistamines onto the bacterial cell surface, expanding the hydrophobicity which, in turn, would increase the surface activity, so antimicrobial response could be due to surface action and due toincreased hydrophobic character Inhibition of efflux pumps,	[35,39,40]
Ceritizine	*Bacillus sp., Vibrio cholera, S. aureus, Escherichia coli and Shigella sp, S. aureus and S. typhi*		
Terfenadine	*MRSA, vancomycin intermediate (VISA), vancomycin-resistant S. aureus, Enterococcus faecium, Enterococcus faecalis, and M. tuberculosis*		
Mepyramine	*Escherichia coli* *, K. pneumoniae*		
**Statins:**			
Atorvastatin, simvastatin, rosuvastatin	*MSSA, MRSA, Εnterococci VSE, Enterococcus VRE, Acinetobacter baumannii, Staphylococcus epidermidis*, *Enterobacter aerogenes, Escherichia coli, Proteus mirabilis, Enterobacter cloacae*	Inhibition of HMG-CoA reductase Cytotoxicity, cells growth suppression, apoptosis promotion	[45,46]

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
