# Peer review of "Antimicrobial Properties on Non-Antibiotic Drugs in the Era of Increased Bacterial Resistance"

_antibiotics, 2020, doi:10.3390/antibiotics9030107_

Round 1
Reviewer 1 Report
The authors reviewed studies investigating antimicrobial properties of non-antibiotic drugs. The manuscript is of relevance and well written.
I have some minor suggestions:
The abstract should be restructured in order to shorten the very long part representing a list of classes of drugs showing antimicrobial properties as well as to include a critical conclusion. Which classes are more promising according to the authors? Which aspects limit the reviewed studies? In order to include a couple of sentences on this aspect, the very long part listing the antimicrobial classes might be shortened. In this sense, the abstract should be more similar to the Introduction, which is currently very short and looks more like an abstract. The authors should briefly described how they selected the reviewed literature. In each section, or at least in a discussion section, I think the authors should add some sentences discussing the critical aspects of investigated literature. For instance, it is not clear which classes of drugs the authors think are more promising or interesting to investigate, which might be promising but interpretation is difficult due to limitations of current literature, which limitations the conducted studies show and so on. In Table 1, I think an additional column listing references of the studies related to each drug class would be usefulAuthor Response
Please see the attachment

Reviewer 2 Report
This short summary of the antimicrobial properties of drugs used for other indications is timely and of use to researchers in the area.
Collecting the info in one place will help researchers gather their thoughts in their own pursuit of antimicrobial compounds.
It would be interesting to include more discussion about the pharmacokinetics and pharmacodynamics of these known and reported compounds and tension this against MIC values - thus the authors can comment upon the details of what kind of dose regimen may be required and if that is feasible or not
Author Response
POINT 1: More discussion about the pharmacokinetics and pharmacodynamics of these known and reported compounds and tension this against MIC values
we have included a lot of known infdormation: pg 3 line 99-107,133-137, pg 5 line 198-206, 234-245 e.t.c
POINT 2:Thus the authors can comment upon the details of what kind of dose regimen may be required and if that is feasible or not
we concluded all the data that already exist

Round 2
Reviewer 1 Report
The authors adequately assessed the raised points.
As a minor comment, please check the revised parts as there are several instances in which two words seem not to be separated by a space.